# Atomic-Scale Understanding of Structure and Properties of Complex Pyrophosphate Crystals by First-Principles Calculations

**Redouane Khaoulaf** [1,2], **Puja Adhikari** [3], **Mohamed Harcharras** [4], **Khalid Brouzi** [5], **Hamid Ez-Zahraouy** [2] **and Wai-Yim Ching** [3,*]

[1] Laboratory of Spectroscopy, Molecular Modeling, Materials, Nanomaterials, Water and Environment, (LS3MN2E-CERNE2D), Faculty of Sciences, Mohammed V University, Av Ibn Battouta, B.P 1014, Rabat 10000, Morocco; rkhaoulaf2000@yahoo.fr

[2] Laboratory of Condensed Matter and Interdisciplinary Sciences (LAMCSCI), Faculty of Sciences, University Mohammed V, Rabat B.P 1014, Morocco; ezahamid@gmail.com

[3] Department of Physics and Astronomy, University of Missouri, Kansas City, MO 64110, USA; paz67@mail.umkc.edu

[4] Laboratory of Materials Engineering and Environment, Université Ibn Tofail, Kenitra B.P 242, Morocco; mharch2009@gmail.com

[5] Energie Matériaux et Développement Durable, EMDD-CERNE2D, Mohammed V University, Rabat B.P 1014, Morocco; khbrouzi@hotmail.com

* Correspondence: Chingw@umkc.edu; Tel.: +1-816-235-2503

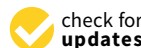

**Featured Application: Pyrophosphate crystals with low effective mass leading to high electron mobility could be used in radiation detectors. In addition, with a large range of absorption in the infrared, visible, and ultraviolet spectral regions, they could be used in the optical field. The prediction of novel mechanical properties of pyrophosphate may lead to new applications.**

**Abstract:** The electronic structure and mechanical and optical properties of five pyrophosphate crystals with very complex structures are studied by first principles density functional theory calculations. The results show the complex interplay of the minor differences in specific local structures and compositions can result in large differences in reactivity and interaction that are rare in other classes of inorganic crystals. These are discussed by dividing the pyrophosphate crystals into three structural units. $H_2P_2O_7$ is the most important and dominating unit in pyrophosphates. The other two are the influential cationic group with metals and water molecules. The strongest P-O bond in $P_2O_5$ is the strongest bond for crystal cohesion, but O-H and N-H bonds also play an important part. Different type of bonding between O and H atoms such as O-H, hydrogen bonding, and bridging bonds are present. Metallic cations such as Mg, Zn, and Cu form octahedral bonding with O. The water molecule provides the unique H···O bonds, and metallic elements can influence the structure and bonding to a certain extent. The two Cu-containing phosphates show the presence of narrow metallic bands near the valence band edge. All this complex bonding affects their physical properties, indicating that fundamental understanding remains an open question.

**Keywords:** Pyrophosphate; electronic structure; mechanical properties; optical properties; first-principles calculations

## 1. Introduction

In comparison with oxides or nitrides, the structure and properties of phosphates are much more complex and less well studied. However, they are critically important and explored in multiple

disciplines such as physics, chemistry, biology, and materials science. In the elementary classification, there are three types of phosphates—monophosphates, condensed phosphates, and oxyphosphates [1]. Phosphoric anion consists of $PO_4$ as a basic unit and is present in any phosphoric anion with P-O-P bonds in the condensed phase. One of the condensed phosphates is the pyrophosphate, also known as diphosphate or dipolyphosphate. This is a subset of family of crystals that originates from pyrophosphoric acid $P_2O_5 \cdot 2H_2O$ ($H_4P_2O_7$) and contains the pentoxide group ($P_2O_5$). Pyrophosphates exist in both crystalline as well as non-crystalline glassy forms. Phosphate glasses are quite different from the silicate-based optical glasses due to low dispersion and high refractive indices. Alkaline earth phosphate glasses are particularly important due to its high transparency in the ultraviolet region [2].

Phosphates have a wide range of scientific and technological applications. Phosphate glasses are used as host materials for rare-earth ions in solid state laser as low temperature sealing glasses [2], and in lithium batteries [3]. They have also been used as source in near-infrared lasers [4]. Phosphate phosphor can be used in field of lighting due to effective excitation in near-ultraviolet range [5], Other uses of phosphate glasses include nuclear power production [6] and piezoelectric material for pressure sensor applications as demonstrate in Ga phosphate crystal [7].

In biomolecular science, phosphates always play a tremendous role. For example, adenosine triphosphate (ATP) and creatine phosphate are the main reservoirs of biochemical energy. Phosphates and especially pyrophosphates are the usual "leaving group" in metabolic reaction [8] i.e., pyrophosphate leaves the reactant, making it more reactive, and this characteristic is used in inhibiting infections [9]. Another example is the use of phosphates in materials, especially calcium orthophosphate, for osteoporosis patients in hip fracture and joint replacement [10] in the human body, which demands special mechanical properties such as high fracture toughness and yield strength. Phosphate-based glasses are increasingly used as the main biomaterials for hard and soft tissue engineering that require a specific biological response [11]. It should also be noted that we actually have a track record in studying the structure and properties of bio-related phosphates including hydroxyapatite (HAp) [12–14] and tricalcium phosphate (TCP) [15].

In addition to biomedical applications, phosphates are also known for applications in several other areas such as energy science, sensors, catalysis, and many more. One of the major applications of phosphates is in agriculture and the development of green energy. They are particularly important in a country like Morocco with a long history of development and natural applications. Indeed, in Morocco, phosphates are exceptionally rich, both in terms of the content of their $P_2O_5$ containing ore and the enormity of their reserves that form the valuable sedimentary series. According to the report published in January 2018 by the United States Geological Survey (USGS), the world phosphate rock reserves are estimated at about 70 billion tons, of which 50 billion tons are in Morocco, which represents three quarters of the world's reserves.

Phosphates are very important for the economy and industrial development in a country like Morocco. The OCP group (Cherifian Phosphates Office) is a Moroccan company created in 1920 that specializes in the extraction, recovery, and marketing of phosphate ($P_2O_5$) and its derived products. It makes Morocco the world's leading market in the production of phosphate products. The treatment of phosphate rocks gives the following marketed products: phosphoric acid ($H_3PO_4$), single superphosphate (SSP), monoammonium phosphate (MAP), diammonium phosphate (DAP), fertilizer (NPK), and triple superphosphate (TSP). The strategy used by the OCP group for the distribution of phosphate ($P_2O_5$) was based on the three sectors, fertilizers (85% by volume of $P_2O_5$), animal nutrition (8%) as well as a wide variety of industrial uses (7%), including laundry detergents and food products such as sodas, etc. Another use is that the purified phosphoric acid is used as salts for the food industry such as yeast, cheese, preservation of meat and fish, treatment of drinking water, etc. Other industrial uses include metal processing, textiles, cements etc.

Given the complex structures of phosphates and their ubiquitous presence, a fundamental understanding of their structures and properties is a subject of paramount importance. However, such studies are lagging far behind other inorganic materials such as silicates. High-level quantum

mechanical calculations always play a role in such cases in providing the information on electronic structure, molecular interaction, and reactivity. They have implications on physical properties including vibrational, optical, and mechanical properties. Such studies on phosphate crystals can facilitate identifying the potential usage of these complex materials. Several such investigations have appeared in the literature in recent years. Tang et al. [16] investigated the behavior of phosphate species using ab initio molecular dynamics. Application of pyrophosphate for water oxidation catalysts was investigated by Kim et al. [17] using both experimental and computational methods. Other density functional theory (DFT)-based calculations for different kinds of pyrophosphates has been done by Witko et al. [18], Zhang et al. [19], and Xiang et al. [20].

In this work, we present the results of calculations of five pyrophosphate crystals including the one that was published a year ago on $K_2Mg(H_2P_2O_7)_2 \cdot 2H_2O$ [21]. The only difference in the structural components of these five crystals are the alkali metal K or $NH_4$ and metallic elements such as Mg, Zn, or Cu. They have the same pyrophosphate group $H_2P_2O_7$ and same number of $H_2O$ molecules. As expected, the electronic structure and bonding in these five crystals are very similar, but to our great surprise, they show some considerable variations. The presence of narrow metallic bands at the top of valence band (VB) in the two Cu-containing pyrophosphates is identified. Also, the mechanical properties with the parameter gauging brittleness and ductility can differ by as much as 50%. Since the mechanical properties of all materials ultimately depends on the strengths of interatomic bonding, what could be the atomistic origin of such unusual behavior for the pyrophosphate crystals? With the detailed and accurate calculations of the electronic structure and mechanical and optical properties, we offer some tangible insights that defies the conventional wisdom. In the next two sections, we briefly describe the crystal structures and computational methods used in the calculation. The main results are presented and discussed in Section 4. The paper ends with a summary and some conclusions.

## 2. Crystal Structures

The five pyrophosphates in this paper are $K_2Mg(H_2P_2O_7)_2 \cdot 2H_2O$, $(NH_4)_2Zn(H_2P_2O_7)_2 \cdot 2H_2O$, $K_2Cu(H_2P_2O_7)_2 \cdot 2H_2O$, $4[K_2Cu(H_2P_2O_7)_2 \cdot 2H_2O]$, and $4[K_2Zn(H_2P_2O_7)_2 \cdot 2H_2O]$. These crystals were synthesized using the wet method. This method is generally applied by the strategy followed by our scientific team (RK, KB, and MH) to obtain multicomponent compounds such as acidic pyrophosphates studied in this work. This method is based on the preparation of a solution of the various precursors, which are generally sodium or potassium pyrophosphates ($Na_4P_2O_7(H_2O)10$, $K_4P_2O_7$), $NH_4Cl$, $MgCl_2$ $(H_2O)_6$, $ZnCl_2(H_2O)_4$, and $CuCl_2(H_2O)_2$. We then resort to the solubilization of all these precursors by the addition of a strong acid of the HCl type (pH of solute must be controlled), sometimes with slow stirring. Afterward, slow evaporation at room temperature for all the solute leads to the formation of single crystals. These pyrophosphates have well-characterized crystal structure as well as vibrational analysis, Raman, and infrared spectra reported such as $K_2Mg(H_2P_2O_7)_2 \cdot 2H_2O$ by Harcharras et al. [22], $(NH_4)_2Zn(H_2P_2O_7)_2 \cdot 2H_2O$ by Capitelli et al. [23], and $4[K_2Zn(H_2P_2O_7)_2 \cdot 2H_2O]$ by Khaoulaf et al. [24]. $K_2Cu(H_2P_2O_7)_2 \cdot 2H_2O$ and $4[K_2Cu (H_2P_2O_7)_2 \cdot 2H_2O]$ are same crystal structure with different space group. i.e., *P*-1 and *Pnma* respectively. Their Raman and infrared vibrational spectroscopic analysis is done by Khaoulaf et al. [25].

The structure of these five crystals after optimization are shown in Figure 1. As already pointed out in ref [21], crystal containing light H atoms have difficulties in accurately locating the positions of H atoms because of the weak intensity signal in the measurement. This results in the unrealistically short O-H bonds. This deficiency has been rectified by accurate ab initio geometry optimization implemented in the Vienna Ab initio Simulation Package (VASP) to be explained in section below. The optimized crystal parameters for all five crystals are shown in Table 1. In the later discussion on the structural characteristics and their physical properties of these five crystals, we will focus on the difference and interplay between the three structural units. The first unit is the cationic group consisting of metallic elements. The only difference in the crystal composition is in the first structural unit K or $NH_4$ and metallic element Mg, Zn, or Cu. The composition from the other two structural units,

pyrophosphate group $H_2P_2O_7$ and water are the same. To save space, we will name these five crystals as C1 to C5 with difference only in the first two components of unit 1: C1 for $K_2Mg(H_2P_2O_7)_2 \cdot 2H_2O$, C2 for $(NH_4)_2Zn(H_2P_2O_7)_2 \cdot 2H_2O$, C3 for $K_2Cu(H_2P_2O_7)_2 \cdot 2H_2O$, C4 for $4[K_2Cu(H_2P_2O_7)_2 \cdot 2H_2O]$, and C5 for $4[K_2Zn(H_2P_2O_7)_2 \cdot 2H_2O]$. It should be pointed out that C3 and C4 have the same chemical composition and formula unit but with different crystal symmetry and volumes. The C1, C2, and C3 crystals have the triclinic structure whereas C4 and C5 crystals have orthorhombic structure and are called composites.

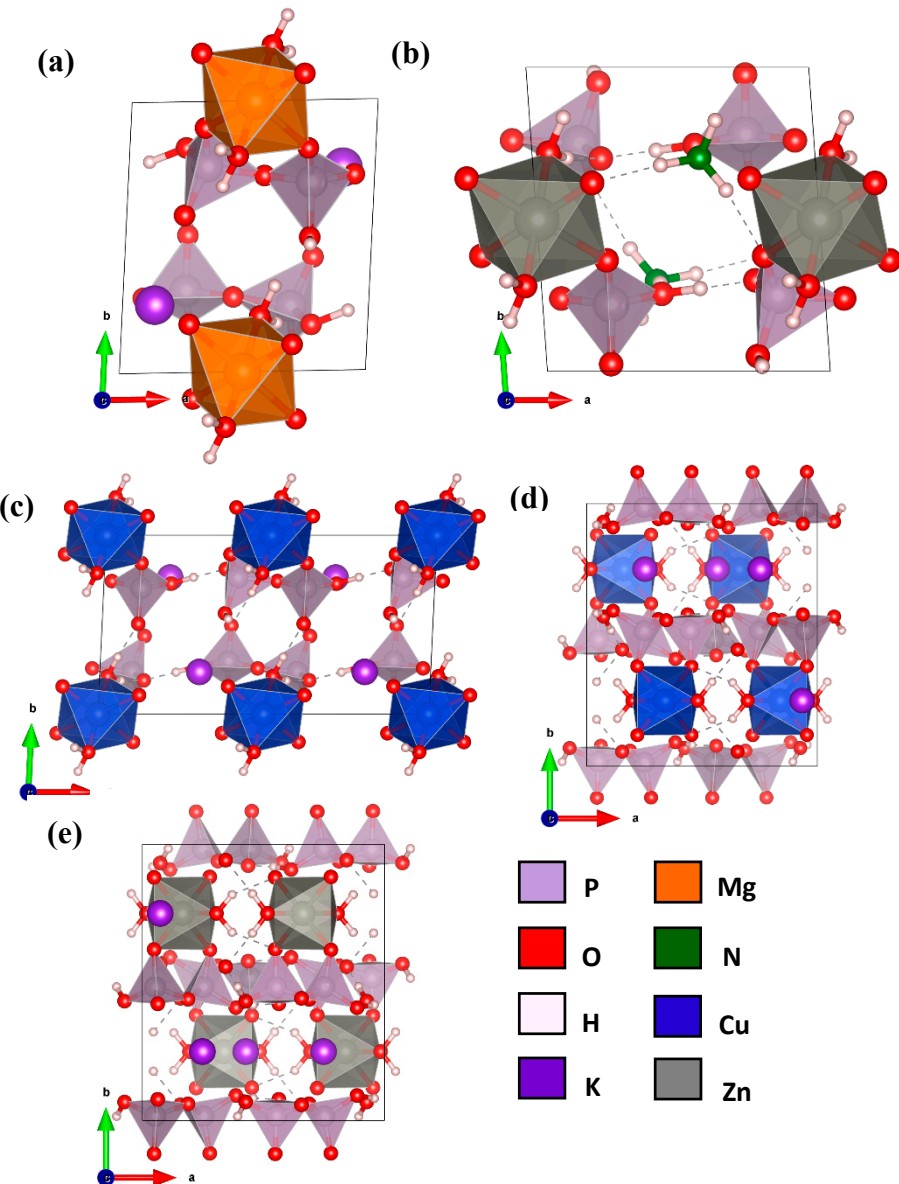

**Figure 1.** Polyhedra figures for the five pyrophosphate crystals viewed along the c-direction. The labels (**a–e**) are $K_2Mg(H_2P_2O_7)_2 \cdot 2H_2O$, $(NH_4)_2Zn(H_2P_2O_7)_2 \cdot 2H_2O$, $K_2Cu(H_2P_2O_7)_2 \cdot 2H_2O$, $4[K_2Cu(H_2P_2O_7)_2 \cdot 2H_2O]$, and $4[K_2Zn(H_2P_2O_7)_2 \cdot 2H_2O]$ respectively.

**Table 1.** Lattice parameters of the five crystals.

|  | Space Group | a(Å), b(Å), c(Å), $\alpha$, $\beta$, $\gamma$ | Volume (Å$^3$) | No. of Atoms |
|---|---|---|---|---|
| **C1** | *P*-1(Ci) | 6.954, 7.503, 7.589, 81.166°, 75.522°, 84.257° | 378.05 | 31 |
| **C2** | *P*-1(Ci) | 7.178, 7.424, 7.808, 81.007°, 71.428°, 90.952° | 388.58 | 39 |
| **C3** | *P*-1(Ci) | 7.101, 7.430, 7.609, 78.761°, 71.657°, 83.958° | 373.38 | 31 |
| **C4** | *Pnma*(D$_{2h}$$^{16}$) | 9.757, 11.134, 13.728, 90.000°, 90.000°, 90.000° | 1491.35 | 124 |
| **C5** | *Pnma*(D$_{2h}$$^{16}$) | 9.770, 11.166, 13.746, 90.000°, 90.000°, 90.000° | 1499.60 | 124 |

## 3. Computational Methods

We used two computational packages in this study: VASP [26] and Orthogonal Linear Combination of Atomic Orbitals (OLCAO) [27]. Both are based on the density functional theory. For structural optimization and elastic properties calculations using VASP, we used the PAW-PBE potential [28] with generalized gradient approximation (GGA) for the exchange correlation potential. We used a relatively high energy cutoff of 600 eV. The electronic and ionic force convergence criteria are set at $10^{-9}$ eV and $10^{-7}$ eV/Å respectively. A $6 \times 6 \times 6$ k-point mesh was used for C1, C2, and C3 crystals and a less dense mesh of $4 \times 4 \times 4$ for the two larger crystals C4 and C5. No discernable difference in the relaxed structure was observed using other exchange correlation functions such as hybrid functional PBE0, HSE03, and applying van der Waals correction in the case of $K_2Mg(H_2P_2O_7)_2 \cdot 2H_2O$.

The calculation of mechanical properties using the optimized structure has been described before [29,30]. Specifically, we apply a small strain $\epsilon$ ($\pm$ 0.25%) to the crystal. The elastic tensor elements are obtained using the stress ($\sigma_j$) vs strain ($\epsilon_i$) response analysis scheme to the relaxed structure and then obtained the elastic coefficients $C_{ij}$ (i, j = 1, 2, 3, 4, 5, 6 (using Hooks law) and compliance coefficient $S_{ij}$ by solving the set of linear equations From the calculated $C_{ij}$ and $S_{ij}$, other mechanical properties such as bulk modulus (K), shear modulus (G), Poisson's ratio ($\eta$), and Young's modulus (E) are obtained using Voight-Reuss-Hill (VRH) polycrystals approximation [31,32].

For electronic structure and interatomic bonding, we use the in-house developed OLCAO with the VASP-relaxed structure as input. A more localized minimal basis (MB) is used for the calculation of effective charge $Q_\alpha^*$ and bond order (BO) values. The BO is the overlap population $\rho_{\alpha\beta}$ between any pair of atoms ($\alpha$, $\beta$) based on Mulliken population analysis [33,34]. Mulliken analysis is only meaningful for comparisons between different materials if the same basis set in the same computational package is used as in the present case. Mulliken analysis is more effective if a localized basis is used, and we always use the minimal basis as implemented in OLCAO method with the same atomic basis for all atomic species [27]. The same approach has been successfully applied to many different materials systems including complex biomolecular systems [35–45].

$$Q_\alpha^* = \sum_i \sum_{m,occ} \sum_{j,\beta} C_{i\alpha}^{*m} C_{j\beta}^m S_{i\alpha,j\beta} \tag{1}$$

$$\rho_{\alpha\beta} = \sum_{m,\ occ} \sum_{i,j} C_{i\alpha}^{*m} C_{j\beta}^m S_{i\alpha,j\beta} \tag{2}$$

In the above equation, $S_{i\alpha,j\beta}$ are the overlap integrals between the $i^{th}$ orbital in $\alpha^{th}$ atom in the $j^{th}$ orbital in $\beta^{th}$ atom. $C_{j\beta}^m$ is the eigenvector coefficients of the $m^{th}$ occupied band. The BO from equation (2) defines the relative strength of the bond. Total bond order density (TBOD) is obtained by normalizing the total BO (TBO) with cell volume. TBOD is a single metric to access the internal cohesion in the crystal and can be decomposed partial components (PBOD) for any structural units or groups of bonded atomic pairs.

For the calculation of interband optical properties in the form of frequency-dependent complex dielectric function $(\hbar\omega) = \varepsilon_1(\hbar\omega) + i\varepsilon_2(\hbar\omega)$, the imaginary part $\varepsilon_2(\hbar\omega)$ is calculated first according to

$$\varepsilon_2(\hbar\omega) = \frac{e^2}{\pi n\omega^2} \int_{BZ} dk^3 \sum_{nl} |\langle \psi_n(k,r)| - i\hbar\nabla |\psi_l(k,r)\rangle|^2 f_l(k)[1 - f_n(k)]\delta[E_n(k) - E_l(k) - \hbar\omega] \quad (3)$$

where $l$ and $n$ are for the occupied and unoccupied states respectively. $\psi_n(k,r)$ are the ab initio Bloch functions from OLCAO calculation using a full basis (FB) set and a larger k-point sampling. $f_l(k)$ and $f_n(k)$ are the Fermi distribution functions. The real part $\varepsilon_1(\hbar\omega)$ is obtained from the imaginary part $\varepsilon_2(\hbar\omega)$ through Kramers-Kronig transformation [46].

The combination of using VASP and OLCAO packages with two different basis expansions is highly effective in revealing the subtle features in the material properties especially on the atomic scale details. The method is particularly useful for highly complex crystals or non-crystalline materials as demonstrated in some of recent publications [35–45].

## 4. Results and Discussion

### 4.1. Electronic Structure

The calculated band structures for the five pyrophosphate crystals are shown in Figure 2. They are all insulators with large direct band gaps at Γ ranging from 3.70 eV to 5.22 eV and listed in Table 3. However, on a closer inspection, we find that in C3 and C4 the states near the VB top are actually metallic bands with both occupied and unoccupied band below and above the Fermi level. This is illustrated in Figure 2c′,d′ next to Figure 2c,d.

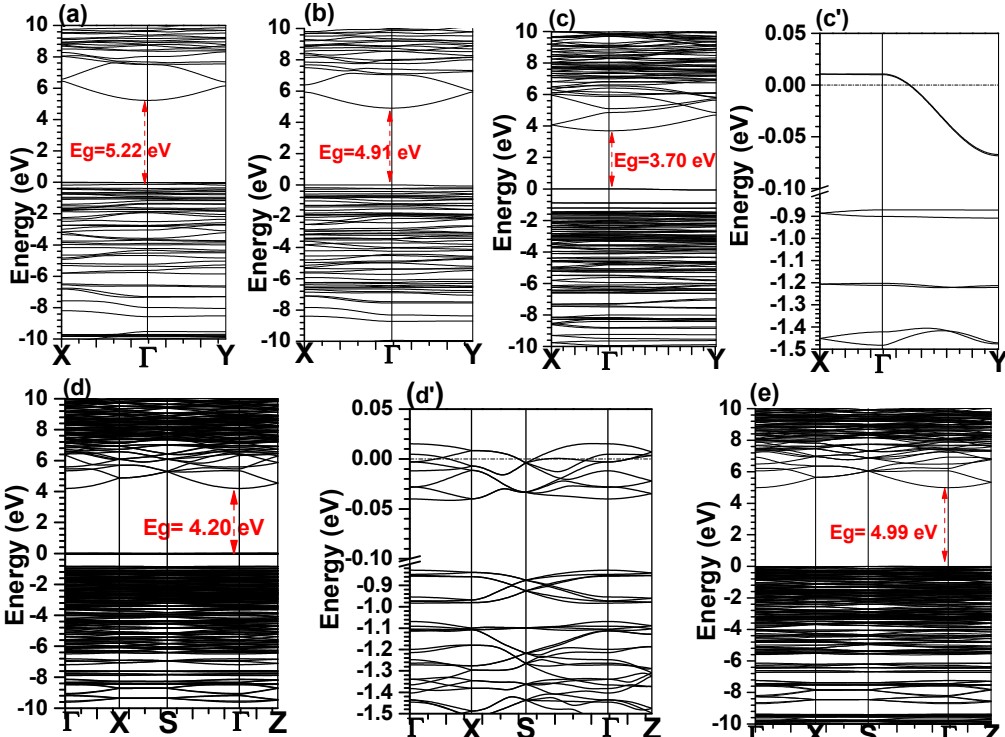

**Figure 2.** Calculated band structures of five pyrophosphate crystals. (**a**) C1, (**b**) C2, (**c**) C3, (**d**) C4, (**e**) C5, and (**c′**), (**d′**) magnified y-axis from −1.5 eV to 0.05 eV for C3 and C4, respectively, with dashed line showing Fermi level.

The top of valence band (VB) in all five crystals are flat and the bottom of conduction band (CB) have curvatures. The calculated CB effective masses listed in Table 1 are comparable with each other

around 0.143 $m_e$ to 0.149 $m_e$. They can be compared to other wide-gap semiconductors such as AIP (0.13 $m_e$) [47], GaN (0.19 $m_e$) [48], ZnSe (0.17 $m_e$) [49], ZnTe (0.16 $m_e$) [50], CdS (0.20 $m_e$) [51], and CdSe (0.13 $m_e$) [52]. The small effective mass implies these crystals could have large electron mobility in the CB. Therefore, pyrophosphate crystals with wide band gap and high electron mobility could be used in radiation detector [53].

The total density of states (TDOS) for all five crystals with energy range from $-25$ to 25 eV are shown in Figure 3. The TDOS are further resolved into partial density of states (PDOS) for the four structural units. These give more detailed information on the interaction between different groups and the difference among five crystals. For example, in crystals containing K, a sharp peak at around-10 eV is due to semi-core nature of K-3p orbital. We see a slight double peak for K due to two different types of K based on sites. $(NH_4)_2Zn(H_2P_2O_7)_2 \cdot 2H_2O$ is the only crystal containing $NH_4$ ligand instead of K. The sharp peak at around $-16.38$ eV is due to both N and H atoms, and a smaller peak at $-6.49$ eV is due to N-2p orbital. For C3 and C4, the gap state arises purely from Cu 3d electrons with possible interaction with water and the pyrophosphate group $(H_2P_2O_7)_2$. In Figure 3f, we specifically show the PDOS of Cu in C3 and C4 in the energy range from $-8$ eV to 8 eV with the zero of energy located at the Fermi level. It should be mentioned that the current calculation is not spin-polarized based on the assumption that the pyrophosphates studied in this paper are all paramagnetic. To get much deeper insight on Cu-containing pyrophosphates, spin-polarized calculation, or DFT +U approach should be attempted.

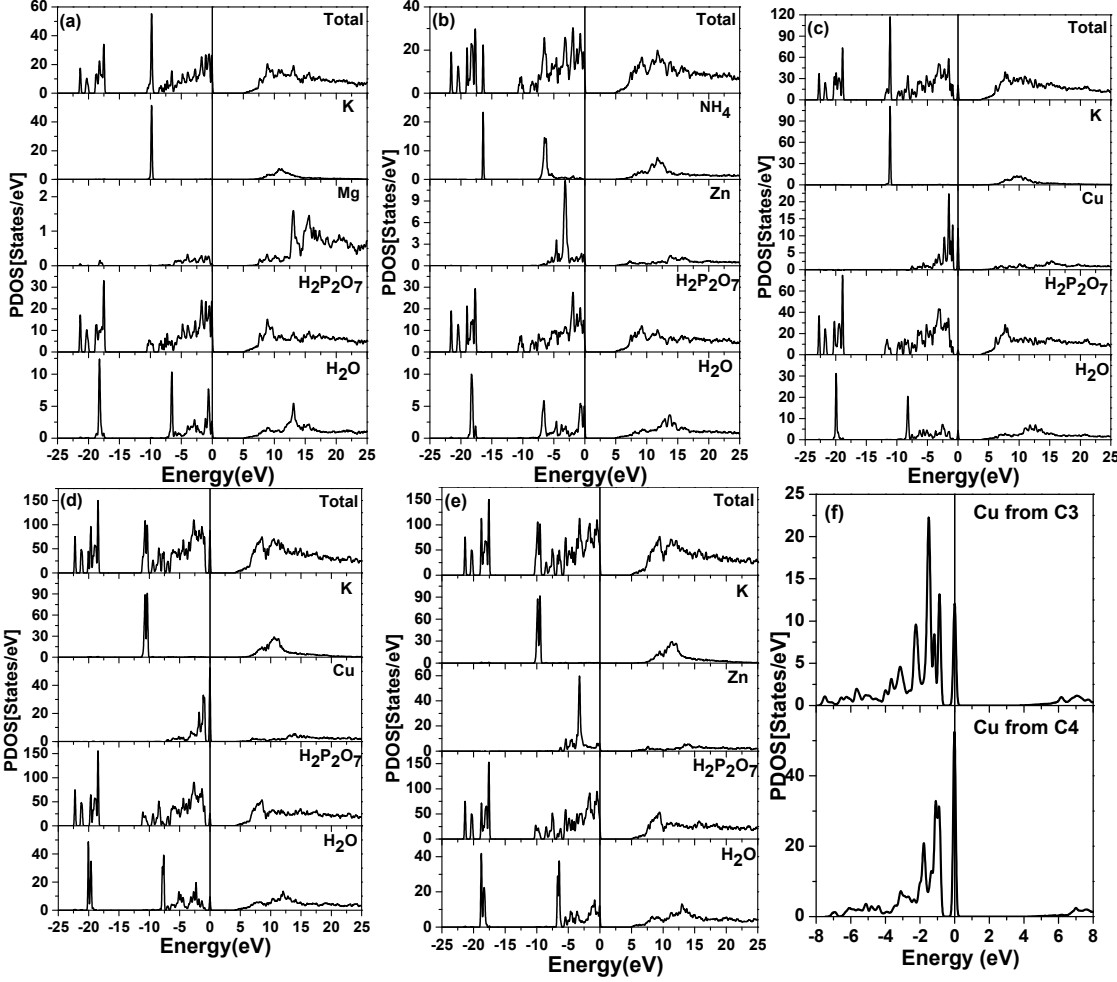

**Figure 3.** Calculated total density of states (DOS) and partial density of states (PDOS) from four structural unit. (**a**) C1, (**b**) C2, (**c**) C3, (**d**) C4, (**e**) C5, and (**f**) shows PDOS of Cu from C3 and C4.

### 4.2. Partial Charge Distribution

The partial charge (PC) $\Delta Q$ on each atom is defined as the deviation of the effective charge $Q_\alpha^*$ (Equation (1)) from the neutral charge $Q^0$ on the same atom, or $\Delta Q = Q^0 - Q_\alpha^*$. We have calculated the PC for every atom in all five crystals as shown in Figure 4. As expected P, Mg, K, Cu, and H have positive PC whereas O and N atoms have negative PC and their values are very similar. Minor variations of the PC for P, H, or O only reflect their locations in the structural units of $H_2P_2O_7$ or $H_2O$. The minor variations can also originate from the different crystal symmetry, triclinic vs orthorhombic. Thus, the PC for each atom in the crystals shown in Figure 4 are not very insightful. On the other hand, grouping the PC for atoms in each of the three structural unit will be quite revealing. They are listed in Table 2. It is noted that there are considerable differences in the PC of the three structural units between the five crystals including the two water molecules ($2 \cdot H_2O$). This indicates that there could be subtle difference in inter molecular interaction and corresponding reactivity.

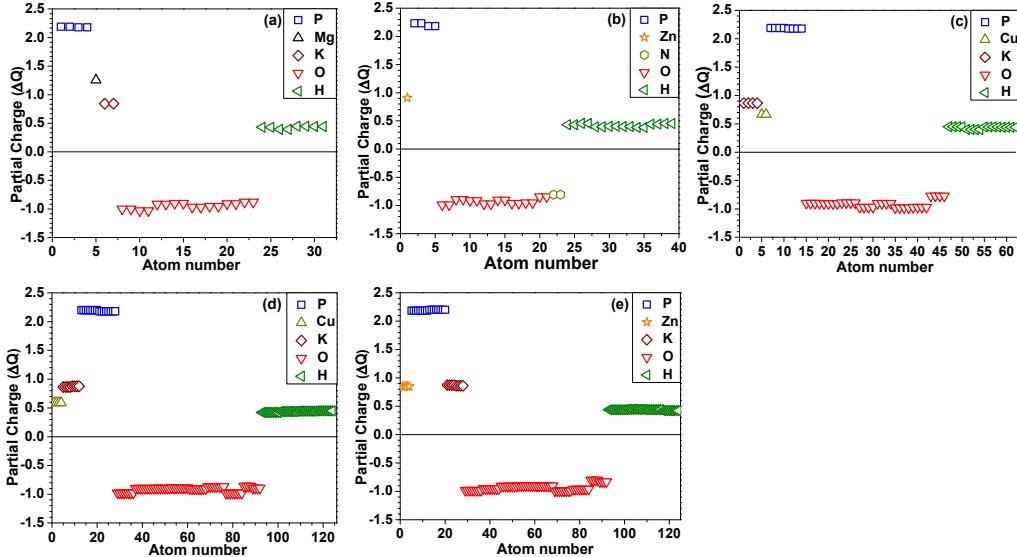

**Figure 4.** Calculated partial charge distribution in five pyrophosphate crystals. (**a**) C1, (**b**) C2, (**c**) C3, (**d**) C4, (**e**) C5.

**Table 2.** Partial charge values from three different structural units in five crystals. The first five columns are the elements in the first structural unit.

|     | $K_2$ | $(NH_4)_2$ | Mg   | Zn   | Cu   | $(H_2P_2O_7)_2$ | $2 \cdot H_2O$ |
| --- | ----- | ---------- | ---- | ---- | ---- | --------------- | -------------- |
| C1  | 1.68  | -          | 1.26 | -    | -    | -2.97           | 0.03           |
| C2  | -     | 1.55       | -    | 0.91 | -    | -2.57           | 0.11           |
| C3  | 1.73  | -          | -    | -    | 0.67 | -2.62           | 0.22           |
| C4  | 1.74  | -          | -    | -    | 0.59 | -2.36           | 0.03           |
| C5  | 1.73  | -          | -    | 0.85 | -    | -2.71           | 0.13           |

### 4.3. Interatomic Bonding

The best way to describe the interatomic interaction is to show the BO vs BL distribution in each crystal and then use the data to obtain the TBOD and PBOD for each structural unit. The BO vs BL distribution for the five pyrophosphate crystals turns out to be quite similar, except for C3 and C4 containing Cu. We present the distribution for crystal C2 ($(NH_4)_2Zn(H_2P_2O_7)_2 \cdot 2H_2O$) and C4 ($4[K_2Cu (H_2P_2O_7)_2 \cdot 2H_2O$) as examples in Figure 5a,b, respectively. More detailed discussion for C1 has been described in [21]. Essentially, there are seven types of different interatomic bonds for each crystal. The five common bonds in each crystal are covalent bonds (P-O, O-H), hydrogen bond (HB)

(O⋯H), bridging bonds (O-H-O), and negligibly weak (O-H) bonds when far apart. The covalent P-O bonds within the structural unit $H_2P_2O_7$ is the strongest bonds, and they are only slightly different in each crystal, which implies the strong tetrahedral unit $PO_4$ as is true in all phosphates. Mg-O and Zn-O bonds are stronger in comparison to K-O bonds, and they form octahedral units in the respective crystal. In C2, the unique covalent N-H bonds are very strong since they are part of the intramolecular bonds in $NH_4$. The BO for Cu-O in C3 and C4 with same chemical composition are quite different (Not shown). In C3, some Cu-O bonds have a larger BL of 2.50 Å with a lower BO of 0.054 e. This shows difference in bonding due to crystal structure. The sum of total bond order values in the crystal when normalized by volume gives the TBOD, a very useful parameter to identify internal cohesion in pyrophosphate crystals, and they are listed in Table 3. It turns out that C2 has the highest cohesion with TBOD of 0.02743 e_/Å$^3$, much larger than the other four crystals with very similar TBOD. The PBOD from different bond types is shown in Figure 5c in the form of histogram. P-O bonds have highest contribution in all crystals and is responsible for its internal cohesion. Apart from the strong N-H bonds in C2, O-H is second most significant bonds in these crystals. However, all other bonds also play its part in crystal cohesion of pyrophosphates including the O⋯H hydrogen bonds.

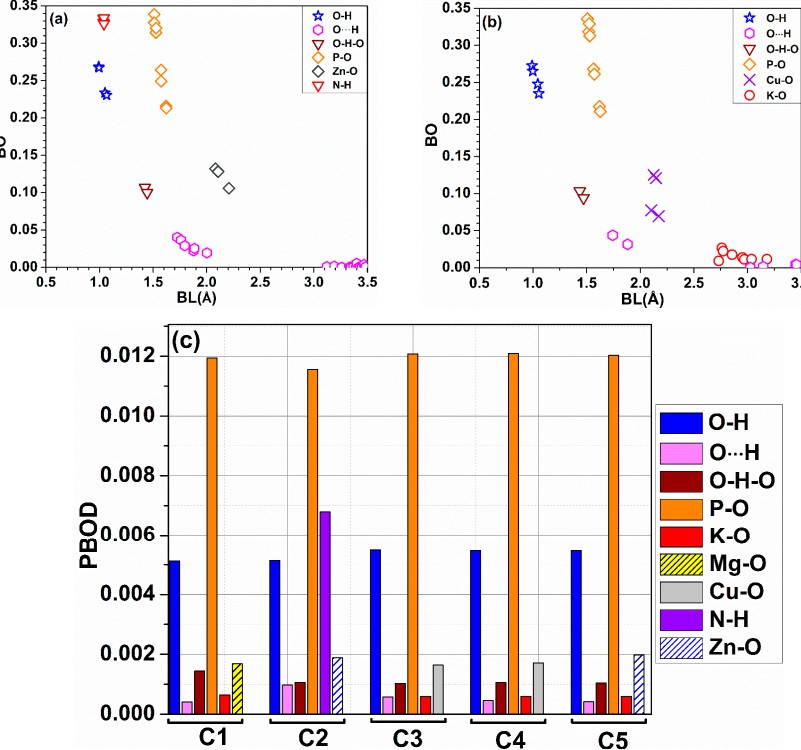

**Figure 5.** (**a**) Bond order (BO) vs. BL for all pairs of atoms in C2 $(NH_4)_2Zn(H_2P_2O_7)_2·2H_2O$, (**b**) C4 $4[K_2Cu(H_2P_2O_7)_2·2H_2O]$, (**c**) bar graph of contributions from different bonds types in the five pyrophosphate crystals.

**Table 3.** Calculated physical properties for the five crystals.

|    | Eg (eV) | $m_e^*$ ($m_e$) | $\varepsilon 1(0)$ | n | $\omega_p$ (eV) | TBOD |
|----|---------|-----------------|--------------------|-----|------------------|---------|
| **C1** | 5.22 | 0.143 | 2.09 | 1.44 | 22.98 | 0.02125 |
| **C2** | 4.91 | 0.146 | 2.11 | 1.45 | 20.85 | 0.02743 |
| **C3** | 3.70 | 0.146 | 2.37 | 1.53 | 23.14 | 0.02141 |
| **C4** | 4.20 | 0.146 | 290.99 | 17.06 | 22.89 | 0.02139 |
| **C5** | 4.99 | 0.149 | 2.15 | 1.47 | 22.67 | 0.02155 |

### 4.4. Mechanical Properties

Mechanical properties are essential for any materials with technological applications. However, there is very little information on the mechanical properties for pyrophosphate crystals in contrast to their vibrational properties. To fill this gap, we have calculated the elastic coefficients from the VASP relaxed structure for the five crystals. From the elastic coefficients, the mechanical parameters for these crystals are obtained using the VRH approximation for poly-crystals [17,18]. They are bulk modulus (K), shear modulus (G), Young's modulus (E), and Poisson's ratio ($\eta$), and are listed in Table 4. Also included are the Pugh ratio G/K and the universal elastic anisotropy parameter $A^U$ [54]. $A^U$ equal to 0 implies perfect isotropy and is usually much less than 2 and seldom goes beyond 4 for most crystals with different crystalline symmetries [54]. There is a large variations of $A^U$ among the five pyrophosphate crystals with a low value of 0.7182 for C4 and high value of 3.610 for C1, thus C1 is far more elastically anisotropic than C4. Generally speaking, low symmetry triclinic crystal should have higher $A^U$ compared to orthorhombic crystals, however, it is still disconcerting that C5, which is also orthorhombic, has much larger $A^U$ than C4. Each of these mechanical parameters are correlated and all are derived from the elastic coefficients. The higher the bulk modulus, the less compressible is the system. Shear modulus is related to the rigidity of the material and Young's modulus represent stiffness of the material. G/K is a parameter based on Pugh's criteria [55] to estimate brittleness or ductility in pure metals from comparative analysis. i.e., if G/K ratio is high, then it is brittle and ductile if the G/K ratio is low. However, Pugh's ratio may also work in other materials, as it also has relation with Poisson's ratio ($\eta$) i.e., higher G/K ratio lower is $\eta$ and vice versa. All five crystals have low K and G values, with K ranges from 22.88 GPa to 30.28 GPa and G ranges from 13.90 GPa to 15.63 GPa respectively. As a result, G/K ratio ranges from a low value of 0.4591for C4 to a much higher value of 0.6831 for C1. This is a variation of almost 50% in the Pugh ratio signal the brittle nature for C1 and being ductile for C4. Given the fact that they have almost identical TBOD (C1: 0.02125 and C4: 0.2139), there must be a credible reason for rationalization with the use of Pugh ratio.

**Table 4.** Calculated Cij and mechanical parameters for the five crystals.

|      | K(GPa) | G(GPa) | E(GPa) | $\eta$ | G/K | $A^U$ |
| --- | --- | --- | --- | --- | --- | --- |
| **C1** | 22.88 | 15.63 | 38.19 | 0.2218 | 0.6831 | 3.6100 |
| **C2** | 30.28 | 14.84 | 38.26 | 0.2894 | 0.4899 | 3.1220 |
| **C3** | 28.19 | 14.66 | 37.48 | 0.2784 | 0.5199 | 2.7776 |
| **C4** | 30.26 | 13.90 | 36.15 | 0.3009 | 0.4591 | 0.7182 |
| **C5** | 24.53 | 15.58 | 38.57 | 0.2379 | 0.6351 | 1.9064 |

In Figure 6, we plot the G vs K in the range for K from 0 to 40 GPa and G from 0 to 20 GPa for the five pyrophosphates C1-C5 together with a selected set of materials within the same range. The slopes of the dashed lines show the G/K ratio which are close to the inverse of the Poisson's ratio for the pyrophosphates. They can be divided into two groups with G/K = 0.611 (C1, C5) and 0.457 (C2, C3, C4). One is ductile and the other is brittle, according to Pugh ratio criterion of greater or larger than G/K around 0.5 [55]. We are not aware of experimental measurement for these pyrophosphate crystals, so the results presented can be considered as theoretical predictions.

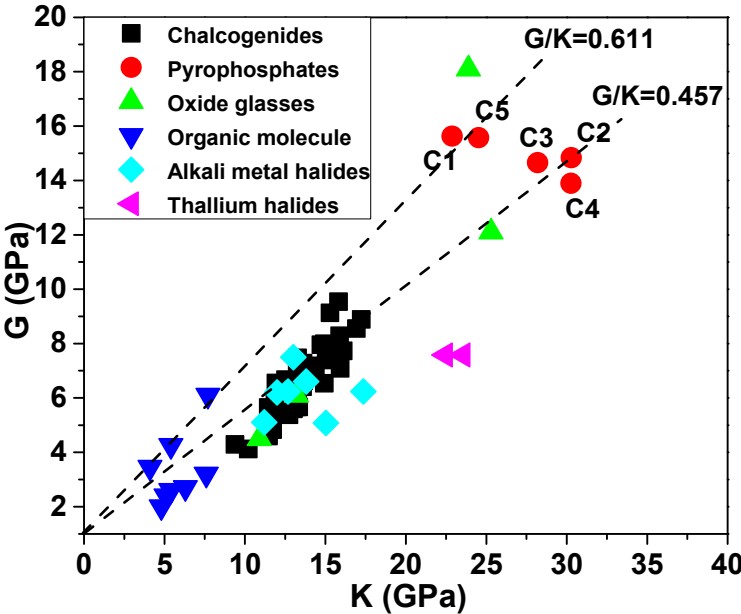

**Figure 6.** Shear modulus G vs bulk modulus K for the 5 pyrophosphates and other selected crystals such as chalcogenides [56], oxide glasses [57], organic molecules [58], alkali metal halides [59,60], thallium halides [60] within the same range of G and K. The slope of the dashed lines gives the G/K values 0.611 and 0.457.

*4.5. Optical Properties*

The optical properties for the five pyrophosphate crystals are calculated in the form of complex dielectric function based on the one-electron theory of interband optical transition. The calculated real ($\varepsilon_1$) and imaginary ($\varepsilon_2$) parts of frequency-dependent dielectric function are shown in black and red color respectively in Figure 7 (center panel (a) to (e)). Optical absorption spectra in all five crystals above the absorption threshold of 5.0 eV show five peaks—A, B, C, D, and E—roughly in the ultraviolet region. This feature is related to the similarity in the TDOS of Figure 3 dominated by the PDOS of the $H_2P_2O_7$ unit. However, for the absorption spectra for the Cu containing crystals C3 and C4 in Figure 7c,d, additional absorptions occur in the 0.0 eV to 5.0 eV region, which are in the infrared, visible, ultraviolet spectral region (see the far left column in Figure 7). This is due to the presence of unoccupied part of the metallic band discussed in Section 4.1. This made the optical properties in C4 resemble a metallic material with huge peaks at energy close to 0.0 eV. On the other hand, the optical properties for C3 are quite reasonable, despite the same chemical composition and formula with C4. The optical absorption ($\varepsilon_2$) in C3 occurs at a higher photon energy than in C4. After Kramers-Kronig transformation of $\varepsilon_2$ to $\varepsilon_1$, $\varepsilon_1$ at zero is only slightly larger than those in C1, C2, and C5 but are still reasonable. A plausible explanation on the difference between C3 and C4 is that they have different crystal symmetry (space group *P*-1(Ci) and *Pnma*($D_{2h}^{16}$), respectively) despite these two Cu-containing pyrophosphates have the same chemical formula. This is one of the very rare examples that the crystal symmetry can play a critical role in the optical properties in the infrared frequency region.

The refractive index for the five pyrophosphate crystals can be obtained from the square root of zero-frequency limit of real part of the dielectric function ($\varepsilon_1(0)$). They are listed in Table 3 and are in the range of 1.44 to 1.53. The value of n = 17.06 for C4 is questionable due to large absorption in the infrared region. These refractive index values can be compared with 60% glucose solution in water (1.44) [61], BaF2 (1.47) [62], CaF2 (1.433) [62], SrF2 (1.44) [62], CsF (1.48) [62], RbBr (1.55) [62], and RbCl (1.49) [62] etc.

In the right panel of Figure 7a'–e', we display the calculated energy loss function (ELF) for the five crystals whose optical absorption spectra are shown in the center panel. The peak of ELF is known as plasma frequency ($\omega_p$), $\omega_p$ is the frequency of collective excitation of electrons in solid at high energy

and can be easily measured experimentally. The $\omega_p$ for all five crystals are listed in Table 3. They range from 20.85 eV in C2 to 23.14 eV in C3.

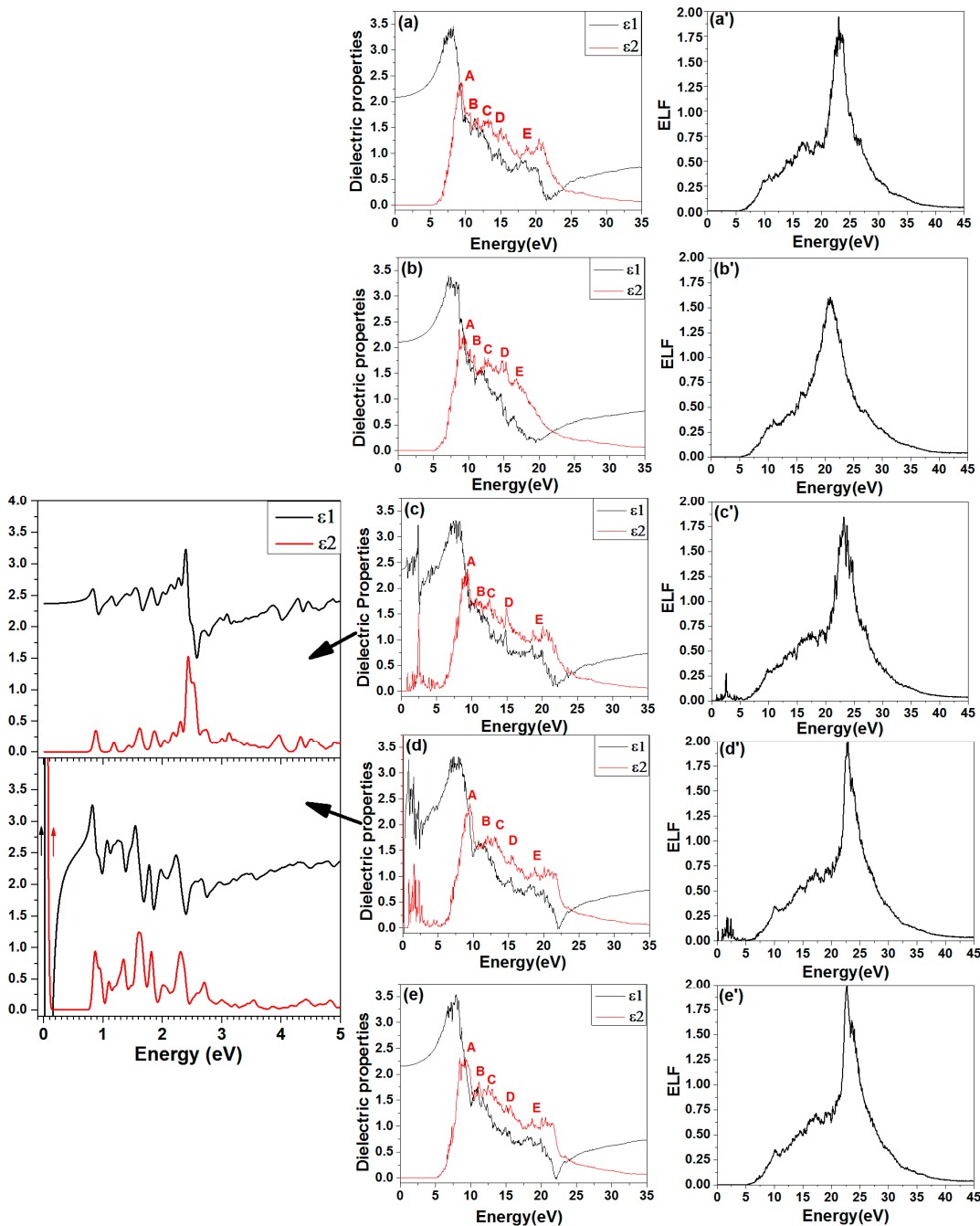

**Figure 7.** Optical properties for the five pyrophosphate crystals: Center panel: real ($\varepsilon_1$) and imaginary ($\varepsilon_2$) parts of the dielectric function. Far left panel: expanded illustration of $\varepsilon_1$ and $\varepsilon_2$ from 0.0 to 5 eV. Right panel, the electron energy loss function (ELF). (**a**,**a'**) for C1; (**b**,**b'**) for C2; (**c**,**c'**) for C3; (**d**,**d'**) for C4; and (**e**,**e'**) for C5.

## 5. Summary and Conclusions

We have presented detailed results on the quantum mechanical calculation on electronic structure, partial charge, interatomic bonding, mechanical and optical properties of five pyrophosphates with very complex structures. To our knowledge, these are the first time such calculations have been attempted. This enable us to make detailed comparisons on the structure and properties among them.

The fundamental understanding of pyrophosphate crystals can open the door for many applications. The general conclusions of this study can be succinctly summarized as follows:

1. The electronic structure and bonding in these crystals appear to be quite similar, but there are subtle differences in bonding in these crystals due to differences in their compositions in different metallic elements and crystal geometry.

2. The presence of very narrow metallic state of Cu in the crystals can induce empty states above the Fermi level, which make its optical properties more complicated and interesting.

3. $NH_4$ is a unique group of atoms to replace the metallic ion. It is the only organic group in the five crystals and has the largest TBOD.

4. Crystal symmetry plays an important role, triclinic vs orthorhombic. This implies short-range intermolecular action is more critical than longer-range crystal symmetry.

5. $H_2P_2O_7$ is the most important and dominating unit in pyrophosphates. The water molecule provides the unique H···O bonds, and metallic elements can influence the structure bonding and reactivity to a certain extent.

6. The mechanical properties of these five crystals vary the most, but detailed correlation to structure and electronic properties remains clear.

The work presented represent the first step in understanding the structure and properties of pyrophosphates. Much work remains to be done, especially in the direction of having different transition metal elements replacing Cu or Zn and invoke spin-polarized calculations for magnetic properties.

**Author Contributions:** W.-Y.C. and R.K. initiated the project. P.A. and W.-Y.C. did the calculations. R.K., M.H., and K.B. developed the synthesis method of phosphate materials including acidic pyrophosphates. All the authors participated in the discussion and interpretation of the results. W.-Y.C., P.A., and R.K. wrote the paper. All authors edited and proof-read the final manuscript.

**Funding:** This research received no external funding.

**Acknowledgments:** This research used the resources of the National Energy Research Scientific Computing Center supported by the DOE under Contract No. DE-AC03-76SF00098 and the Research Computing Support Services (RCSS) of the University of Missouri System. PA is supported by a research grant from the School of Graduate Studies at UMKC.

**Conflicts of Interest:** The author declares no conflict of interest.

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
