# Peer review of "Atomic-Scale Understanding of Structure and Properties of Complex Pyrophosphate Crystals by First-Principles Calculations"

_applsci, doi:10.3390/app9050840_

Round 1

Reviewer 1 Report

The paper presents the results of calculations of five pyrophosphate crystals and discusses the electronic structure and mechanical and optical properties. This is a carefully done study and the findings are of considerable interest. My detailed comments are as follows:

1. Although this paper is good, so much calculation has been done. it would be ever better if some deeply explanations of the results were added.

2. In Figure 1, these five structures are not clearly enough, especially the (c), (d) and (e). It`s better to repaint the 3 picture.

3. The figures are not in a same style, it makes the paper seems chaotic.

4. As for the electronic structure,to account better for strong correlations in the partially filled d or f shell, the DFT+U method self-interaction corrected DFT scheme are considered.

5. The conclusion should be concise and only summarize the most important contribution of the research.

Author Response

We thank the referee 1 for useful comment. They are responded below.

We agree with the opinion that there could be more deep explanations to some of the results. This will be done in future publications since some of these can be very subtle. 

We improved the figure 1 to make (a)-(f) all consistent with a slight change in the Fig. caption.

We have improved figure 1 as mentioned above.

The referee is correct that in metals with d and f electrons, DFT+U will be important to account for the band gap. In the present study, only Cu and Zn are involved. They have full shell of 3d electrons and as such the effect of U is negligible. 

We follow the referee's suggestion and made the Conclusion more focused and succinct.   

Reviewer 2 Report

In this work, the authors present a theoretical study by first-principle methods of five pyrophosphate crystal structures. These structures exist because they have been synthetized and characterized by experiment.
This study provides information about electronic structure, mechanical and optical properties of these materials and suggests possible use in practical applications. This work is complementary with previous experimental studies, it is competently written and provides an exhaustive bibliography on the topic.
I recommend publication in Applied Sciences in the present form.
~                                                                     

Author Response

We appreciate reviewer's very positive assessment of our work. 

Reviewer 3 Report

This first-principles study reports results on five different pyrophosphate crystals. Overall the manuscript is in good shape, however there are some unclear scientific issues which should be addressed satisfactorily. I also noticed carelessness in some sentences. I suggest the authors read their manuscript carefully before submitting to avoid simple typing mistakes. I provide a list for both types of concerns. I recommend revision subject to further review.

Scientific issues:

* Band structure, Fig. 2 and 3:

It is not clear where the Fermi level is located at. I assume at 0 eV. This should be clearly stated. It is crucial because in the case of Cu containing compounds the authors are talking about a "gap state" which is half-occupied. If that state has nonzero occupation then the Fermi energy should be located at the energy of this state (at about 1 eV)! By definition, no state with energy higher than Fermi energy can be occupied in a first-principles calculation (0 Kelvin situation). Since this problem occurs for the Cu case, I think the first-principles calculations for these two compounds need additional care. First possibility to check is spin polarization. Authors do not mention anything about the formal or effective valence state of Cu, but if it is +1, ie. a d9 configuration, then this "half-occupancy" situation can be solved by developing spin magnetic moments at Cu sites. The second possibility, which is more applicable, is that the situation resembles the high-Tc Cu-O superconductors. In cuprates the LDA or GGA treatments yield metallic DOS contrary to the band gaps observed in experiments, and only upon introducing an on-site Coulomb repulsion, the Hubbard U parameter for the 3d electrons of Cu, one can obtain reasonable electronic structure. Here, if I understand the term half-occupied correctly, we actually have metallic DOS in the Cu compounds, ie. these Cu compounds are not insulators per se. Therefore, I strongly recommend performing DFT+U calculations to understand the Cu compounds properly. At the present stage the reported results are misleading and cannot be trusted.

* Optical properties, Table 3:

The refractive index for C4, one of the Cu compounds, is not trustable, as the authors themselves state. Something must be wrong with that calculation. The other Cu compound has reasonable values, so perhaps the problem is that related to the "half-occupied" Cu d band located around 1 eV. Nevertheless, one should not publish results that s/he does not believe. If there is an unexpected result, then the authors should be able to explain how and why. If the reasons are not clear even to them, then there is no point in trying to publish those results. I suggest doing these calculations with another code to cross check. In any case, the DFT+U method should be tried first.

*Mechanical properties, ductility vs brittleness issue:

The authors' numerical results based on elastic constants calculations are very clear and consistent: C1 and C5 are brittle, others are ductile according to both the Pugh criterion and the Poisson's ratio criterion. Therefore I see no problems here. The problem must be the authors' way of interpreting the bond order density. The bond order calculation based on Mulliken-type idea is in general not a reliable procedure. Mulliken analysis depends sensitively on the basis sets used, and even the direction of charge transfer can be obtained wrong, for example, by a Mulliken population analysis, As a result we have two issues in regard to bond orders: (i) their calculated values may be questionable, (ii) the connection between bond order density and mechanical properties is not quantitatively well established. On the other hand, Pugh's and Poisson's ratio criteria correlate quite well with experimental data. Hence, the sentence on lines 280 and 281 should be removed. the discussion should be modified.

* Misc.

line 222: "neutral charge" what do you mean by that? Neutral charge of any atom is zero. Do you mean the atomic number Z, ie. the number of electrons in the free atom? Please clarify.

Table 2: The partial charge values have too many digits. Are you sure that all those 4 digits after the decimal point are converged, ie. they will not change if you repeat the calculation with more k-points or with larger basis sets, etc.? Significant digit concept applies to computational studies, as well, it is not only for experimentalists.

line 249: It may be a good idea to calculate cohesive energies and compare them with the TBOD values. Such a comparison may enhance the value of the sentence on this line.

Language related:

line 64: are increasingly used

line 109: differing

Fig.1 caption: add respectively to the end.

line 135: used --> implemented in the

line 139: consisting

line 157: correction the in case of --> correction in the case of

line 168: "values. The bond order (BO) values. The BO .." : Remove the words between the fullstops, or complete them to a sentence.

line 187: f1(k) --> fl(k) (ie. use l the letter instead of number 1)

line 191: the word atomic between in and the material does not look correct, probably a leftover from a correction. Rewrite the sentence correctly.

line 215: only the --> the only

line 246: Rewrite the sentence "The BO .."  The "in C3 than in C4" part is not correct.

line 262: "modulus (K), ..." which modulus? I guess bulk, but it should be written there explicitly.

line 264 and 265: The sentence on 264 and what is written on line 265 do not make any sense. Probably something is missing in between. Please rewrite these parts in an understandable way.

Author Response

Dear Editor of Applied Sciences and reviewer 3,

We appreciate the comments and suggestions of the additional reviewer on our manuscript entitled “Atomic-scale Understanding of Structure and Properties of Complex Pyrophosphate Crystals by First-principles Calculations”. Based on these comments, we have thoroughly revised the manuscript. We want to thank this expert reviewer for his/her time in carefully reading our manuscript providing very constructive suggestion which we will addressed below point by point. The reviewer’s comments are listed as italics for easy reference. The corresponding modifications or additions to the text are underlined.

Scientific issues:

(1) Band structure, Fig. 2 and 3:

It is not clear where the Fermi level is located at. I assume at 0 eV. This should be clearly stated. It is crucial because in the case of Cu containing compounds the authors are talking about a "gap state" which is half-occupied. If that state has nonzero occupation then the Fermi energy should be located at the energy of this state (at about 1 eV)! By definition, no state with energy higher than Fermi energy can be occupied in a first-principles calculation (0 Kelvin situation). Since this problem occurs for the Cu case, I think the first-principles calculations for these two compounds need additional care. First possibility to check is spin polarization. Authors do not mention anything about the formal or effective valence state of Cu, but if it is +1, ie. a d9 configuration, then this "half-occupancy" situation can be solved by developing spin magnetic moments at Cu sites. The second possibility, which is more applicable, is that the situation resembles the high-Tc Cu-O superconductors. In cuprates the LDA or GGA treatments yield metallic DOS contrary to the band gaps observed in experiments, and only upon introducing an on-site Coulomb repulsion, the Hubbard U parameter for the 3d electrons of Cu, one can obtain reasonable electronic structure. Here, if I understand the term half-occupied correctly, we actually have metallic DOS in the Cu compounds, ie. these Cu compounds are not insulators per se. Therefore, I strongly recommend performing DFT+U calculations to understand the Cu compounds properly. At the present stage the reported results are misleading and cannot be trusted.

Response:

Our definitive for 0.0 eV is always to be the top of the occupied valence band for insulators where a band gap exist. This is energy is obtained by averaging the top of the occupied energy levels over the symmetric directions in the Brillouin zone. This energy is then assign as the 0.0 eV with all other energies shifted by this averaged energy. This is certainly straight forward in the case for C1, C2 and C5 crystals which are insulator. For metallic systems, 0.0 eV is assigned to be the Fermi level of the metal by convention. In our original submission,  we noticed that for copper containing pyrophosphates C3 and C4, a sharp peak exist within the range of an apparent gap similar to C1, C2 and C5 crystals and we erroneously assumed it to be a ½ occupied gap state in order to interpret  they optical absorption spectra in later section. Upon careful examination of the bands near in this peak and to our great surprise, this “peak” actually consists of an extremely narrow metallic band and they have both occupied and unoccupied bands below and above the Fermi level (See Fig. 2(f) in the revised manuscript).  The situation is very similar to YBCO higher temperature superconductor except the unoccupied metallic band in YBCO is much wider, about 2 eV) (see, W.Y. Ching et al, PRL, 59, 1333, 1987). We sincerely apologize for this careless oversight. We want to thank the referee for the careful and insightful scrutiny that leads us to the discovery this error.

The present calculation is non-spin polarized on the assumption that these phosphates are all paramagnetic insulators with full 3d10 (not d9) shell for Cu and Zn but with possible interaction with other atoms in the crystal to introduce a defect-like very narrow metallic band in the case of C3 and C4. This is partly support by the partial charge calculation (Table) for Cu to be +0.25e/Cu (0.55/4) due to charge transfer from the 3s electron of Cu (neutral electron configuration for Cu is 3d104s1).The referee is absolutely correct that DFT + U calculation may be able to further clarify this point. However, in the present work, the crystals are large and complex, such calculation is a non-trivial task that could take us a year to complete. The main focus for this paper is the comparison between five pyrophosphate crystals on their optical and mechanical properties. Moreover, the precise value of U to be used for Cu is not known especially in a complex pyrophosphates. In this regards, we feel that spin-polarized DFT + U calculation on two of the five crystals is beyond the scope of current work. Our first-step is to investigate the electronic structure and properties of complex pyrophosphates with the presence of water molecules and competing bonding among difference atomic species. We will plan to work in the more rigorous direction in our next step for phosphates containing some transition metals.    

Actions taken:

We have added the figure 2(c’) and 2(d’) next to Figure 2 (c) and (2d) to show the existence of the very narrow metallic bands and the Fermi level (0.0 eV) in C3 and C4.

1. In the abstract, we have added the following sentence.

The two Cu-containing phosphates show the presence of narrow metallic bands near the valence band edge.

2. Additional comments are added on line 109-110 which read as follows.

“The presence of narrow metallic bands at the top of the valence band (VB) in the two Cu-containing pyrophosphates is identified.

3. Additional comments are added on line 2xx-2xx which read as follows.

However, on a closer inspection, we find that in C3 and C4 the states near the VB top are actually metallic bands with both occupied and unoccupied band below and above the Fermi level. This is illustrated in Fig. 2(c’) and 2(d’) next to Fig. 2(c) and 2(d).

4. On line 2xx-2xx, we specifically added the following in line of referee suggestions, that the current calculation is only a first step, a more rigorous spin-polarized calculation or DFT + U approach will be a better alternative.

In Fig.3 (f), we specifically show the PDOS of Cu in C3 and C4 in the energy range from -8 eV to 8 eV with the zero of energy located at the Fermi level. It should be mentioned that the current calculation is not spin-polarized based on the assumption that the pyrophosphates studied in this paper are all paramagnetic. To get much deeper insight on Cu-containing pyrophosphates, spin-polarized calculation or FDT +U approach should be attempted.

(2) Optical properties, Table 3:

The refractive index for C4, one of the Cu compounds, is not trustable, as the authors themselves state. Something must be wrong with that calculation. The other Cu compound has reasonable values, so perhaps the problem is that related to the "half-occupied" Cu d band located around 1 eV. Nevertheless, one should not publish results that s/he does not believe. If there is an unexpected result, then the authors should be able to explain how and why. If the reasons are not clear even to them, then there is no point in trying to publish those results. I suggest doing these calculations with another code to cross check. In any case, the DFT+U method should be tried first.

Response:

We fully agree with the referee that the refractive index for C4 is not trustable. This is an artifact related to optical transitions associated with the Cu-induced metallic band in C4 discussed above This made its optical properties in C4 resemble a metallic material with huge peaks at energy close to 0.0 eV. On the other hand, the optical properties for C3 is quite reasonable despite the same chemical composition and formula with C4. The optical absorption (ε2) in C3 occur at a higher photon energy than in C4.  After Kramers-Kronig transformation of ε2 to ε1, ε1 is only slightly larger than those in C1, C2 and C5 but still reasonable. A plausible explanation on the difference between C3 and C4 is they have different crystal symmetry (space group P1(Ci) and Pnma(D2h16) respectively) despite these two Cu-containing pyrophosphate have the same chemical formula. This one of the very rare example that the crystal symmetry can play a critical role in the optical properties in the infrared frequency region.

Actions taken:

We have added the additional figure for the optical properties in the 0.0 to 5 eV region of photon energy on the left column of Fig.7 to show the metallic behavior in C4 and its difference with C3.

Additional comments are added on line 3xx-3xx read as follows.

However, for the absorption spectra for the Cu containing crystals C3 and C4 in Fig. 7(c) and 7 (d), additional absorptions occur in the 0.0 eV to 5 eV region which are in the infrared, visible, ultraviolet spectral region. (See the far left column in Fig. 7)  This is due to the presence of unoccupied part of the metallic band discussed in Section 4.1. This made its optical properties in C4 resemble a metallic material with huge peaks at energy close to 0.0 eV. On the other hand, the optical properties for C3 is quite reasonable despite the same chemical composition and formula with C4. The optical absorption (ε2) in C3 occur at a higher photon energy than in C4.  After Kramers-Kronig transformation of ε2 to ε1, ε1 is only slightly larger than those in C1, C2 and C5 but still reasonable. A plausible explanation on the difference between C3 and C4 is they have different crystal symmetry (space group P-1(Ci) and Pnma(D2h16) respectively) despite these two Cu-containing pyrophosphate have the same chemical formula. This one of the very rare example that the crystal symmetry can play a critical role in the optical properties in the infrared frequency region.

(3) Mechanical properties, ductility vs brittleness issue:

The authors' numerical results based on elastic constants calculations are very clear and consistent: C1 and C5 are brittle, others are ductile according to both the Pugh criterion and the Poisson's ratio criterion. Therefore I see no problems here. The problem must be the authors' way of interpreting the bond order density. The bond order calculation based on Mulliken-type idea is in general not a reliable procedure. Mulliken analysis depends sensitively on the basis sets used, and even the direction of charge transfer can be obtained wrong, for example, by a Mulliken population analysis, As a result we have two issues in regard to bond orders: (i) their calculated values may be questionable, (ii) the connection between bond order density and mechanical properties is not quantitatively well established. On the other hand, Pugh's and Poisson's ratio criteria correlate quite well with experimental data. Hence, the sentence on lines 280 and 281 should be removed. the discussion should be modified.

Response:

We appreciate referee’s positive comments on our calculation of mechanical properties of pyrophosphates which is again the first time for such calculations. For the two issues raised:

(a) We agree that Mulliken analysis depends sensitively on the basis sets used and may not be sufficiently accurate. Some other methods such as Bader’s topological scheme or those based on Gaussian package for detailed charge distribution analysis may be more accurate. However, these methods are restricted to simpler crystals or small molecules where the structure of the crystal and their topology are well defined. For complex crystals such as pyrophosphates in this paper, a dense three dimensional grid and numerical evaluation of the charge densities at each grid point is a daunting if not impossible task. In all ab initio DFT calculations, it is always a balancing act between efficiency and accuracy.  We have now stressed that Mulliken analysis is only meaningful for comparisons between different materials if the same basis set in the same computational package is used as in the present case.    Mulliken analysis is more effective is a localized basis is used and we always use the minimal basis in all our calculations as implemented in OLCAO method with the same atomic basis for all atomic species. The same approach has been successfully applied to many different materials systems including complex biomolecular systems (see ref. 36-46). So it is well tested and the calculated values for the intended purpose should not be questionable.

(b) We also agree that the connection between the total bond order density (TBOD) (not bond order density) and the mechanical properties of a material is not well established so far. TBOD is a single quantum mechanical metric to characterize the internal cohesion of a crystal. It is the sum of all bond order values of the crystal normalized by its volume. It does not depend on any geometric parameters and can be effectively divide into PBOD from different types of bond in the crystal (see Fig.5) providing very useful insight on the relative strength of the bond in a multicomponent crystal. In this paper, we show that the TBOD for the five pyrophosphate crystals are quite similar with C2 slightly larger. The bulk modulus for C2 is also the largest among the five crystals but its correlation with other parameters are less clear. We are still in search for a better correlation between mechanical properties and TBOD and PBOD. In other materials systems with larger samples. If firmly established, TBOD can be a valuable descriptor in machine learning and data mining protocol in materials research.

Actions taken:

(a)/We have added the following comments to reflect the above opinion on line 175-179 which read as follows.

  “Mulliken analysis is only meaningful for comparisons between different materials if the same basis set in the same computational package is used as in the present case. Mulliken analysis is more effective is a localized basis is used and we always use the minimal basis as implemented in OLCAO method with the same atomic basis for all atomic species [27]. The same approach has been successfully applied to many different materials systems including complex biomolecular systems (see ref. 36-46).”

(4) Misc.

line 222: "neutral charge" what do you mean by that? Neutral charge of any atom is zero. Do you mean the atomic number Z, ie. the number of electrons in the free atom? Please clarify.

Table 2: The partial charge values have too many digits. Are you sure that all those 4 digits after the decimal point are converged, ie. they will not change if you repeat the calculation with more k-points or with larger basis sets, etc.? Significant digit concept applies to computational studies, as well, it is not only for experimentalists.

line 249: It may be a good idea to calculate cohesive energies and compare them with the TBOD values. Such a comparison may enhance the value of the sentence on this line.

Language related:

line 64: are increasingly used

line 109: differing

Fig.1 caption: add respectively to the end.

line 135: used --> implemented in the

line 139: consisting

line 157: correction the in case of --> correction in the case of

line 168: "values. The bond order (BO) values. The BO .." : Remove the words between the full stops, or complete them to a sentence.

line 187: f1(k) --> fl(k) (ie. use l the letter instead of number 1)

line 191: the word atomic between in and the material does not look correct, probably a leftover from a correction. Rewrite the sentence correctly.

line 215: only the --> the only

line 246: Rewrite the sentence "The BO .."  The "in C3 than in C4" part is not correct.

line 262: "modulus (K), ..." which modulus? I guess bulk, but it should be written there explicitly.

line 264 and 265: The sentence on 264 and what is written on line 265 do not make any sense. Probably something is missing in between. Please rewrite these parts in an understandable way.

Response and Actions taken:

All the above listed errors of all corrected or changed as can be seen from the tracking in the revised version. We really appreciate the careful reading by the referee and we apologize for some careless mistakes and errors. We also modified the conclusion slightly to reflect the changes made in this revision. We end the conclusion with the following sentences. “The work presented represent the first step in understanding the structure and properties of pyrophosphates.  Much work remains to be done especially in the direction of having different transition metal elements replacing Cu or Zn and invoke spin-polarized calculations for magnetic properties.

In summary, we have appropriately responded all the issues raised by the referee. All his/her suggestions are either adopted or explained in the revised manuscript and show in the edited tracking. Three figures are now added to Fig. 2, Fig. 3 and Fig. 7 to support the revisions made in response to the referee. Although, we are unable to comply with the request to do additional calculations using more advance approach since the techniques used in computation will be very time consuming but will not affect our main conclusions, and is beyond the scope of the current work. We look forward to your favorable decision of this paper which was invited by Professor Sergio F. Sousa, the Guest Editors for the Special Issue: The Application of Quantum Mechanics in the Reactivity of Molecules in the Journal Applied Sciences.

Sincerely yours,

Wai-Yim Ching,

Curator’s Distinguished Professor of Physics and Corresponding author.

Round 2

Reviewer 3 Report

The revised manuscript satisfies most of the concerns mentioned in the review. The unsatisfied issues involve the additional calculations needed to address the correlated nature of the states at the top of the Fermi energy for the C3 and C4. I understand that DFT+U calculations may be quite time consuming (since one should also play with different values of U, etc.). I think I will accept the solution offered by the authors, that they state the problem and mention what should be done for a better or more involved treatment of those correlated Cu states.I hope the authors will find the time to examine C3 and C4 with DFT+U method and report their results in future. I recommend acceptance.

I have a suggestion for the Conclusions section: The authors mention spin-polarized calculations for future work. I think the really important issue is the electron correlations. Therefore, the necessity of DFT+U calculations in future work should be emphasized. The DOS figures for C3 and C4 in the revised version are very clear: we do have correlated electrons.  Hence, it is also clear how to go further with a more correct treatment of the electronic structure.

Another comment on C3 and C4: It was very nice of the authors that they presented the detailed bands around 0 eV for C3 and C4. I think the difference between C3 and C4 in terms of the optical properties lies in the following: C4 has 4 times more Cu atoms than C3. Thus, C3 has one band crossing the Fermi energy, but C4 has 8 bands (if I counted correctly) in the close vicinity of the Fermi energy. This situation with many occupied and unoccupied bands with very small energy differences may cause numerical problems in the optics calculations yielding very large numbers at the end. C3 case would be expected to be OK because there is only one band around 0 eV, others are farther away. Anyways, something to think about ...